# Nutritional Adequacy and Day-to-Day Energy Variability: Impacts on Outcomes in Severe Trauma Patients

**DOI:** 10.3390/nu17193180

**Published:** 2025-10-09

**Authors:** Jovana Stanisavljevic, Nikola N. Grubor, Sergej Marjanovic, Ivan Palibrk, Mihailo Bezmarevic, Jelena Velickovic, Adi Hadzibegovic, Marija Milenkovic, Sanja Ratkovic, Bojan Jovanovic

**Affiliations:** 1Department of Anesthesiology, Emergency Center, University Clinical Center of Serbia, 11000 Belgrade, Serbiabjovanovic@outlook.com (B.J.); 2Faculty of Medicine, University of Belgrade, 11000 Belgrade, Serbiaivanpalibrk@yahoo.com (I.P.); lelavelickovic@gmail.com (J.V.); 3Institute for Medical Statistics and Informatics, Faculty of Medicine, University of Belgrade, 11000 Belgrade, Serbia; nikola.n.grubor@med.bg.ac.rs; 4Department of Anesthesiology, Clinic for Digestive Surgery, First Surgical Hospital, University Clinical Center of Serbia, 11000 Belgrade, Serbia; 5Department of Hepatobiliary and Pancreatic Surgery, Clinic for General Surgery, Military Medical Academy, University of Defense, 11042 Belgrade, Serbia; bezmarevicm@gmail.com

**Keywords:** clinical nutrition, severe trauma, caloric adequacy, protein adequacy, ICU, outcomes

## Abstract

Background: Optimal energy and protein delivery during the early phase of severe trauma remains unclear. Observational studies frequently contradict the findings of randomized controlled trials, raising concerns about confounding factors. The aim of this study is to assess nutritional adequacy and daily variability in the energy gaps and its impact on outcomes using innovative statistical methods. Methods: Prospective observational study enrolled severely injured patients in the ICU at the Level 1 trauma center between October 2023 to April 2025. To describe the evolution of calorie and protein deficits during the first 10-day ICU stay, we utilized a linear mixed-effects model to estimate each patient’s individual energy gap trajectory. Results: 286 patients were analyzed. Median APACHE II and ISS score was 16.0 (12.0–20.0) and 22.0 (18.0–27.0), respectively. Mortality rate was 35.3%. Patients received 68.3% of prescribed calories and 76.8% of proteins. Admission energy deficit, rate of caloric intake, and their interaction are associated with ICU mortality. Increased day-to-day energy variability was associated with longer duration of mechanical ventilation (HR = 0.55, 95% CI: 0.31–0.99; *p* = 0.047). Patients who achieved better caloric (HR = 0.68, 95% CI: 0.48–0.98, *p* = 0.036) and protein (HR = 0.29, 95% CI: 0.09–0.96, *p* = 0.043) nutrition had a lower hazard of developing nosocomial infection. Conclusions: This study supports the 2023 ESPEN guidelines, showing that achieving the recommended energy and protein intake during the early phase of severe trauma is linked to lower mortality rates, shorter mechanical ventilation time, and reduced risk of nosocomial infections.

## 1. Introduction

Severe trauma patients are particularly vulnerable to metabolic and immune dysfunction in the early phase of injury, which is associated with an increased incidence of nosocomial infections, prolonged ICU stays, and significant ICU mortality [1]. Medical nutrition extends beyond preventing malnutrition and micronutrient deficiencies providing metabolic optimization, maintenance of gut integrity, prevention of bacterial translocation, attenuation of insulin resistance, and modulation of the immune response [2]. Although these effects are proven to decrease undesirable outcomes, randomized controlled trials (RCTs) and observational studies provide conflicting evidence regarding the optimal amounts of calories and protein requirements in the early phase of critical illness to im-prove outcomes [3]. It is evident that a “one-size-fits-all” nutritional approach can lead to both underfeeding and overfeeding, resulting in unpredictable outcomes and increased mortality. Several studies in trauma populations have concluded that predictive equations recommended in critical care do not reliably estimate energy needs in this group, highlighting that indirect calorimetry (IC) remains the preferred method for determining energy requirements [4,5]. While a meta-analysis of eight RCTs indicated a potential benefit of IC-guided energy delivery on short-term mortality in critically ill patients, the variability in results suggests that further investigation is needed to fully understand its impact [5].

Protective nutritional strategies aim to prevent overfeeding during this vulnerable early phase, as evidence suggests that autophagy and ketogenesis positively affect mitochondrial hibernation [2,3,6,7,8,9]. This presents particular challenges for sedated patients, who may inadvertently receive excess calories from propofol. Additionally, existing research is limited by protocol variability and a tendency toward aggressive early nutrition, often neglecting the benefits of protective strategies [10,11].

To address this gap, our study follows the 2023 guidelines from the European Society for Clinical Nutrition and Metabolism (ESPEN) for clinical nutrition in the ICU by progressively increasing energy and protein intake and utilizing IC measurements [10]. The aim of this study is to evaluate the effects of energy and protein adequacy during the early phase of severe trauma on outcomes, including mortality, incidence of infections, and duration of mechanical ventilation.

## 2. Materials and Methods

A prospective observational study of consecutive severe trauma patients (Injury Severity Score [ISS] > 15) admitted to two ICUs of a Level 1 trauma center was conducted from October 2023 to April 2025. Patients under 18 and those with burn injuries were excluded from the study. The flowchart of included patients is shown in Figure 1. To reduce heterogeneity related to very short ICU stays and to focus on patients in whom nutritional support is most relevant, the main analysis was restricted to those who remained in the ICU and required nutritional support for at least 10 days. Patients discharged or transferred earlier generally had shorter critical care needs and were less likely to require prolonged or targeted nutrition. Based on this criterion, 342 patients with severe trauma were not included in the main analysis due to an ICU stay of <10 days.

### 2.1. Data Collection

Demographic and clinical data included patient age, gender and Body Mass Index (BMI). The burden of comorbid conditions was presented using the Charlson Comorbidity Index (CCI). Injury characteristics encompassed the ISS score [12,13] severity of injury, damage control surgery, and the number of surgeries. Disease severity, calculated over the first 24 h of ICU admission, was assessed using the Acute Physiology and Chronic Health Evaluation II (APACHE II) score and Sequential Organ Failure Assessment (SOFA) score, while nutritional status was evaluated with the Nutrition Risk in Critically ill (NUTRIC) score [14].

### 2.2. Nutritional Data and Calculations

Feeding details, including timing, route (enteral or parenteral), dose and composition of nutrition, reasons for feeding interruption, and complications were prospectively recorded for the first 10 nutritional days. The first nutritional day was uniformly defined as starting at 9:00 a.m. the day after ICU admission. All study patients received commercial nutritional formulas, so the precise amounts of calories and protein provided were measured. Total calorie intake was calculated by combining energy from enteral nutrition (EN), parenteral nutrition (PN), and non-nutritional calories (including glucose solution, propofol, and citrate).

Following ESPEN guidelines [10], we prioritized early EN. In the absence of contraindications, early EN was initiated via a gastric or jejunal route, using a standardized polymeric isocaloric formula (1 mL ≈ 1 kcal), followed by a transition to a high-energy, high-protein formula, with feeding rates increased as tolerated. EN was delivered as a continuous infusion over 24 h, with the infusion rate adjusted to meet energy and protein targets while ensuring tolerance. The decision to switch to enteral boluses was at the clinician’s discretion, guided by ongoing assessments of patient tolerance and nutritional requirements. PN is defined as the intravenous administration of a combination of at least two macronutrient components (lipids, amino acids, and/or glucose), whereas administering glucose alone, regardless of concentration, does not qualify as parenteral nutrition [15]. Supplemental PN is used when EN was insufficient to meet the patient’s nutritional needs or when EN was contraindicated.

Energy requirements were initially estimated using a weight-based equation: 10 kcal/kg on day 1, 15 kcal/kg on day 2, and 20 kcal/kg on day 3. From day 4 to day 10, energy expenditure was measured daily via indirect calorimetry (IC) in mechanically ventilated patients, whenever possible, with the goal of providing 100% of measured energy expenditure [16]. These measurements were performed using a gas exchange module (E-sCOVX) connected to a ventilator (GE Carescape R860, GE Healthcare), following a 20 min stabilization period with no physical activity [17]. For practical reasons, IC was deferred during the initial 72 h to prioritize patient stabilization [16,17,18]. When IC was not feasible to perform, a fixed energy target was used: 25 kcal/kg from day 4 to day 7, and 30 kcal/kg from day 7 to day 10. For the calculations, we used the body mass measured upon admission with a calibrated electronic bed scale, when available, along with any relevant history or information obtained from the patients’ families. Based on ESPEN guidelines, our protein intake was gradually increased during the initial days of treatment, with the target set at 0.5 g/kg on day 1, 0.7 g/kg on day 2, and 1.0 g/kg on day 3. From day 4 to day 7, the target was 1.3 g/kg. Starting from day 7 the target was 2 g/kg of protein, following ASPEN guidelines for trauma patients. It is important to note that, especially for patients with traumatic brain injury (TBI), protein should constitute approximately 15–20% of total caloric intake, often requiring at least 2 g/kg/day, as supported by ASPEN guidelines. Our practices were aligned with these recommendations, with adjustments made based on clinical judgment and evolving needs [19]. A low-protein diet (0.7 g/kg/d) was used for patients with chronic kidney disease or acute kidney injury who were not receiving renal replacement therapy [20]. However, if the BMI was greater than 30 kg/m^2^, we used adjusted body mass for energy and protein calculation [10].

### 2.3. Outcome Variables

Patients were followed for at least 90 days or until discharge if hospitalization exceeded 90 days, or until death. The primary outcome data was ICU mortality. Secondary outcomes included mechanical ventilation (MV) days and the development of nosocomial infection [21].

### 2.4. Statistical Analysis

To describe how patients’ nutritional intake changed during the first 10 days in the ICU, we used a linear mixed-effects model to estimate each patient’s energy gap trajectory, defined as the daily difference between prescribed and actual caloric intake. The model included fixed effects for age, SOFA, APACHE II, weight, height, and ICU Day, allowing each patient to have their starting value (intercept) and trend over time (slope). We then used these individual intercepts and slopes, representing early underfeeding and the pace of nutritional improvement, as predictors in models examining their association with ICU outcomes. This two-part approach allowed us to summarize how each patient’s nutrition evolved and relate that summary to clinical events. We assessed the influence of nutritional intake, intake variability, and protein intake on several outcomes: hospital LOS, MV duration, time to ICU-acquired infection, and ICU mortality. After model fitting we visualized the relationship between the main nutritional variables of interest with their respective 95% confidence intervals and interrogated the relationship by testing specific hypotheses of no mortality difference between specific nutritional states. Intake variability is measured using the root mean square of successive differences (RMSSD), a time-domain statistic which is a measure of variability in short-term fluctuations of any continuous measurement series. RMSSD allows us to capture sudden fluctuations in feeding due to surgery and other interruptions. We also calculate and present the mean absolute successive differences (MASD) metric, which is more robust to extreme variation compared to simple standard deviation for completeness. To account for competing risks, we used Fine and Gray regression [22], which estimates the likelihood of an event over time while considering the possibility of alternative outcomes. To improve interpretability and comparability, continuous nutritional predictors were standardized to have a mean of 0 and a standard deviation of 1 before inclusion in models. The analysis included nutritional predictors such as standardized slope and intercept of the energy gap trajectory, energy intake, protein intake, intake variability (RMSSD), and duration of supplemental PN, along with clinical covariates (age, sex, SOFA, APACHE II, ISS, CCI, feeding route, early EN, MV duration). The influence of protein intake was modeled separately to explore its independent relationship with each outcome while accounting for overall caloric intake, in line with best practices to avoid the “Table 2 Fallacy” [23], the mistaken interpretation of adjusted variables as independent exposures. Sensitivity analysis was conducted using a joint longitudinal-survival model to check if the nutritional associations from the main mortality analysis are maintained. All models were implemented in R version 4.4.2. Mixed-effects models were fit using the lme4 package, and competing risk models were estimated using the cmprsk package.

## 3. Results

### 3.1. Study Population

The demographic and clinical characteristics of the patients are summarized in Table 1.

### 3.2. Calorie and Protein Intake

The nutritional regimens of the patients are shown in Table 2. Only 28.0% of patients received EN in first 48 h. Indirect calorimetry was used to estimate energy requirements for 38.6% of the 2860 nutritional days assessed. Over the first 10 days, patients received an average of 1227.5 kcal/day, while their required intake was 1835.8 kcal/day, meaning they received 68.3% of their prescribed calories. For protein, patients needed an average 105.1 g/day but only received 79.0 g/day, which is 76.8% of their prescribed protein. Figure 2 shows that the provision of calories and proteins progressively increased over the first 10 days. By day 10, 74.8% of patients achieved >70% of prescribed calories and 68.9% achieved >70% of prescribed protein; 52.4% met both targets. Most patients received combined enteral and parenteral nutrition (EN + PN; 92.3%), while 7.7% received EN alone; PN alone and absence of nutritional support were not observed. The average proportion of caloric intake was 63.7% via EN and 33.1% via PN, giving an EN:PN ratio of approximately 2:1. Shock requiring high doses of vasopressors (54 patients, 18.9%) and bowel resection surgery (25 patients, 8.7%) were the most common reasons for not initiating early EN. In 130 patients (45.5%), daily interruptions in feeding were primarily caused by definitive trauma surgeries; other reasons were gastrointestinal (GI) intolerance in 46 patients (16.0%) and diagnostic procedures performed outside the ICU in 40 patients (14.0%). Delays in starting oral feeding were attributed to factors such as weaning from MV, extubation, and the presence of delirium (65 patients, 22.7%). The most common GI complication in trauma patients was paralytic ileus (215 patients, 75.2%), with a median time to first defecation of 7 days (2–15), followed by GI intolerance presented with high gastric residual volumes (GRV), along with diarrhea as another frequent complication.
nutrients-17-03180-t002_Table 2Table 2Nutritional regimen characteristics at 10-day patient averages (*n* = 286).**Initiation of EN (hours)**
<48 h, *n* (%)80 (28)>48–72 h, *n* (%)77 (26.9)>72 h, *n* (%)129 (45.1)**Type of Nutrition**
EN + PN, *n* (%)264 (92)EN only, *n* (%)22 (8)**Caloric and Protein Adequacy Assessment**
Received kcal/day1227.5 *±* 387.9Required kcal/day1835.8 *±* 335.8% Caloric adequacy68.3 *±* 22.8Received protein (g)/day79.0 *±* 23.8Required protein (g)/day105.1 *±* 20.4% Protein adequacy76.8 *±* 22.7% EN of total kcal63.7 *±* 22.5% PN of total kcal33.1 *±* 20.5**Reasons for Delay in Starting EEN**
Shock requiring high dose of vasopressors, *n* (%)54 (19)Bowel resection, *n* (%)25 (8.7)**Reasons for Interrupting Feeds**
Any surgery, *n* (%)130 (45)Weaning/extubation/delirium, *n* (%)65 (23)GI intolerance/high GRV, *n* (%)46 (16)Diagnostic procedures outside ICU, *n* (%)40 (14)**Complications**
Paralytic ileus, *n* (%)215 (75)Time to first defecation, median (Q1, Q3) ^a^7 (2, 15)Diarrhea, *n* (%)46 (16)**Method Used to Estimate Energy Requirements (Number of Measurements) ^b^**Indirect calorimetry, *n* (%)1105 (39)Weight-based, *n* (%)1755 (61)Data are presented as mean ± standard deviation (SD), unless otherwise indicated. EN, enteral nutrition; PN, parenteral nutrition; GRV, gastric residual volume. ^a^ In the first 10 days; ^b^ N = 2860 nutritional days.

### 3.3. Clinical Outcomes

#### 3.3.1. Nutrition and Mortality

Individual energy gap trajectories stratified by 10-day ICU mortality are shown in Figure 3. Marginal effect analysis of the multivariable model showed that an increase in the slope of the daily energy gap from 200 to 0 kcal/day (that is, from a steep daily worsening to no worsening) was associated with a significant reduction in the probability of ICU mortality (absolute change in probability = −0.23, 95% CI: −0.41 to −0.05, *p* = 0.014; Figure 4B). Sensitivity analysis using a joint longitudinal survival model showed that this association is robust and is maintained (HR 0.05, 95% CI 0.001–0.71, *p* < 0.001). This suggests that progressive worsening of energy delivery over time is associated with higher risk of death. A higher initial energy gap (0 vs. −500 kcal/day at ICU admission) was not significantly associated with mortality (*p* = 0.19, Figure 4A), nor was the protein deficit (*p* = 0.46, Figure 4E). However, patients in the highest quartile of nutritional adequacy had a significantly lower mortality probability compared to those in the lowest quartile (absolute change in probability = −0.24, 95% CI: −0.32 to −0.16, *p* < 0.001, Figure 4D), indicating a robust protective association of meeting energy targets.

The predicted marginal effects of the generalized additive model demonstrated distinct non-linear associations between energy intake dynamics and ICU mortality Figure 4 and Figure 5.

#### 3.3.2. Nutrition and Mechanical Ventilation

In mechanically ventilated ICU patients, several factors influenced the duration of MV (see Table A1). Greater day-to-day energy variability (higher RMSSD) was associated with significantly delayed time to extubation (HR = 0.55, 95% CI: 0.31–0.99; *p* = 0.047), as was a larger initial energy deficit (HR = 0.60, 95% CI: 0.40–0.90; *p* = 0.014). A faster recovery of the energy deficit not significantly related to extubation (HR = 0.73, 95% CI: 0.52–1.03; *p* = 0.077). Higher GCS were strongly associated with earlier extubation (HR = 1.10 per point, 95% CI: 1.03–1.17; *p* = 0.004), while increasing age (HR = 0.98 per year, *p* = 0.005) and higher ISS (HR = 0.90, 95% CI: 0.86–0.93, *p* < 0.001) were linked to prolonged MV. Calorie and protein intake, early EN, and comorbidity burden were not independently associated with duration of MV.

#### 3.3.3. Nutrition and Nosocomial Infection

Patients who achieved better nutritional adequacy had a lower hazard of developing nosocomial infection (HR = 0.68, 95% CI: 0.48–0.98, *p* = 0.036; see Table A2). After adjusting for total caloric intake, higher protein intake was also independently associated with a lower risk of nosocomial infection (HR = 0.29, 95% CI: 0.09–0.96, *p* = 0.043). Among clinical covariates, SOFA score at admission, age, sex, APACHE II score, Charlson Comorbidity Index, and injury severity were not significantly associated with risk of nosocomial infection.

## 4. Discussion

To the best of our knowledge, this study is the first to evaluate the impact of nutritional adequacy in the early phase of severe trauma on outcomes, employing innovative statistical methods.

The results of our study reveal a high ICU and 90-day mortality rate of 35.3%. We specifically excluded patients who stayed in the ICU for less than 10 days to focus on a cohort with more critical conditions, which may significantly impact mortality. This can explain the stark contrast between our findings and the adjusted mortality rates reported in recent American trauma studies (12% to 13%) and European studies (7% to 14%) for patients with an ISS greater than 15 [24,25]. Additionally, Japan reported a national average mortality rate of 21.3% for severely injured patients, which further highlights the regional differences in trauma outcomes [4,24,25,26]. In our study, the median APACHE II score was 16 and the median SOFA score was 6, indicating high severity of illness at admission. Moreover, 29.0% of patients had a NUTRIC score ≥ 5, which has been linked in prior studies to higher mortality risk [14,26,27]. It is well-established that ICU mortality is closely linked to nosocomial infections; 60.1% of our patients developed infections, a rate slightly higher than reported in other ICUs [28].

Despite a protocolized nutrition prescription, only 28.0% of patients received any calories within the first 48 h and 64.9% by 72 h. Delayed initiation of early EN and frequent feeding interruptions—primarily due to hemodynamic instability, surgery, airway procedures, and GI intolerance—were the main contributors to inadequate nutritional delivery. These findings are consistent with prior reports in critically ill, particularly trauma, populations [29]. The median calorie intake was 15.2 kcal/kg/day, closely matching the 15.9 kcal/kg/day over the first 15 days after admission reported in the largest prospective European ICU study. Protein provision in our cohort was substantially higher (0.98 vs. 0.7 g/kg/day), a finding consistent with a cohort of severe trauma patients that reported 0.7 g/kg/day [30,31]. We achieved this higher protein intake by switching from an isocaloric to a high energy, high protein enteral formula and progressively titrating feed rates as tolerated; when supplemental PN was required, intravenous glutamine was added. Non-nutritional calories, mainly from propofol infusion, contributed minimally: they declined from 10.5% of total calorie intake on day 1 to <0.4% by days 7–10.

Similarly to Zusman et al., our results suggest that the initial energy deficit has a U-shaped association with mortality risk, and the relationship between protein intake and clinical outcomes follows a comparable pattern [32]. Both marked caloric deficit and excess increased predicted mortality, with lowest risk at 70–100% of caloric targets, highlighting the value of near-target feeding. However, it is essential to note that our mean early intake (15 kcal/kg/day) falls in the mildly hypocaloric range (8–16 kcal/kg/day; 30–60% of target) according to older meta-analyses, which either showed no mortality effect [33,34] or reduced mortality [35,36].

Marginal effects analysis indicated that while the initial energy gap at admission and protein deficit were not significantly associated with mortality, a progressive decline in daily energy delivery of even 200 kcal/day was harmful. It can be appreciated that patients are much more tolerant of early underfeeding than overfeeding (Figure 4A) and that mortality risk rises with extremes of day-to-day variability, but once patients are adequately fed they appear more tolerant to variation (i.e., feeding interruptions). These findings suggest that steady correction of deficits, avoidance of under- and overfeeding, and maintenance of adequate intake with moderate variability may represent clinically relevant strategies which can be developed [6,7,8].

Multiple studies in severe trauma reports marked nitrogen losses, suggesting protein intakes >1.5 g/kg/day, together with immune enhancing diets, may be required to re-store balance [28,37,38]. Although we did not measure nitrogen balance or directly test immunonutrition, our nutritional formulas contained glutamine, arginine, nucleotides, and omega 3. Importantly, greater protein adequacy in our trauma cohort was significantly associated with fewer nosocomial infections—a result that contrasts with RCTs in critically ill patients reporting no outcome benefit and possible reductions in quality of life from high protein immune modulating nutrition [39,40,41].

Compared with the EDEN trial—which evaluated patients with acute lung injury and found no benefit of trophic vs. full enteral feeding on ventilator free days, our cohort (predominantly mechanically ventilated for chest and brain injury) showed that larger initial energy deficits and greater day to day energy variability were significantly associated with prolonged MV [42]. These findings suggest that early consistency and adequacy of energy delivery may influence duration of MV.

We observed that supplemental PN markedly increased nutritional adequacy during EN interruptions: over 90% of patients received combined EN + PN (EN:PN ≈ 2:1), and more than half achieved adequate nutrition. Although the 2023 European ICU guidelines do not support routine early PN in trauma patients [10], supplemental PN has been shown to improve cumulative energy balance and reduce infectious morbidity [6,43], a finding consistent with our observation that negative cumulative energy balance independently increased infection risk.

Overall, research on calorie and protein delivery in trauma patients is limited as severely injured individuals make up only a small percentage of participants in large studies conducted in general ICUs. Calorie deficits and outcomes in trauma patients are similar to those seen in trials of critically ill populations. However, the effects of protein adequacy on infection rates in severe trauma cases are slightly different. This difference indicates a need for further investigation.

### Limitations and Strengths

This study provides valuable insights as it is the first to focus on nutritional therapy in critically ill trauma patients conducted in our country. However, it has several limitations, including a small sample size, single-center design, observational nature, and inconsistent use of IC. This variability in the use of IC may lead to inconsistent estimates of energy requirements, which could influence the calculated energy gaps. Nevertheless, reporting IC use is important because it demonstrates the practical feasibility of implementing this method in trauma patients. In this study, we included only patients who stayed in the ICU for more than 10 days to effectively evaluate the impact of nutritional therapy during their ICU stay. Patients with shorter stays were most often transferred to other units and lost to follow-up, limiting reliable outcome ascertainment. Consequently, our study population primarily represents high-risk trauma patients with prolonged ICU stays, and we acknowledge this limits generalizability to less severely ill populations, in whom nutrition is likely to play a smaller role. Prior evidence suggests nutritional effects are more likely to become detectable after 5 days of controlled nutrition; by focusing on patients beyond 10 days, we maximized our ability to observe effects on mortality while reducing bias from incomplete follow-up. However, this exclusion criterion introduces selection bias and may overlook early deaths and rapid recoveries, potentially leading to overestimation of both mortality rates and the apparent effect of nutrition on outcomes. The employed statistical approach has notable strengths, such as the use of patient-specific linear mixed-effects models to capture individualized calorie deficit trajectories and competing risk regression to account for discharge and mortality as competing events, allowing for a dynamic description of the mortality risk curve. Comprehensive covariate adjustment and standardization of nutritional predictors improved comparability and interpretability. However, assumptions of linearity and proportional hazards may not be fully met, residual confounding is possible, and uncertainty increases at the extremes of estimated slopes and intercepts due to the rarity of observed outcomes at those extremes.

## 5. Conclusions

This study supports the 2023 ESPEN guidelines and the use of IC in severe trauma patients. Providing nearly the target calories is beneficial for reducing mortality. PN successfully filled the gaps when EN was interrupted, improving overall nutritional intake. Lower day-to-day energy variability and deficits were linked to earlier weaning from MV, and better protein intake independently reduced the risk of hospital infections.

## Figures and Tables

**Figure 1 nutrients-17-03180-f001:**
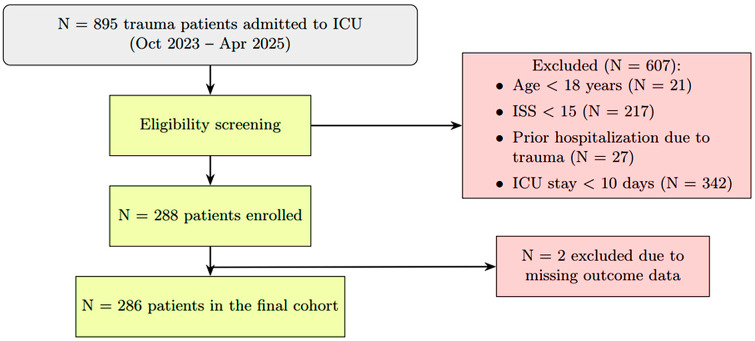
Flowchart of patient enrollment.

**Figure 2 nutrients-17-03180-f002:**
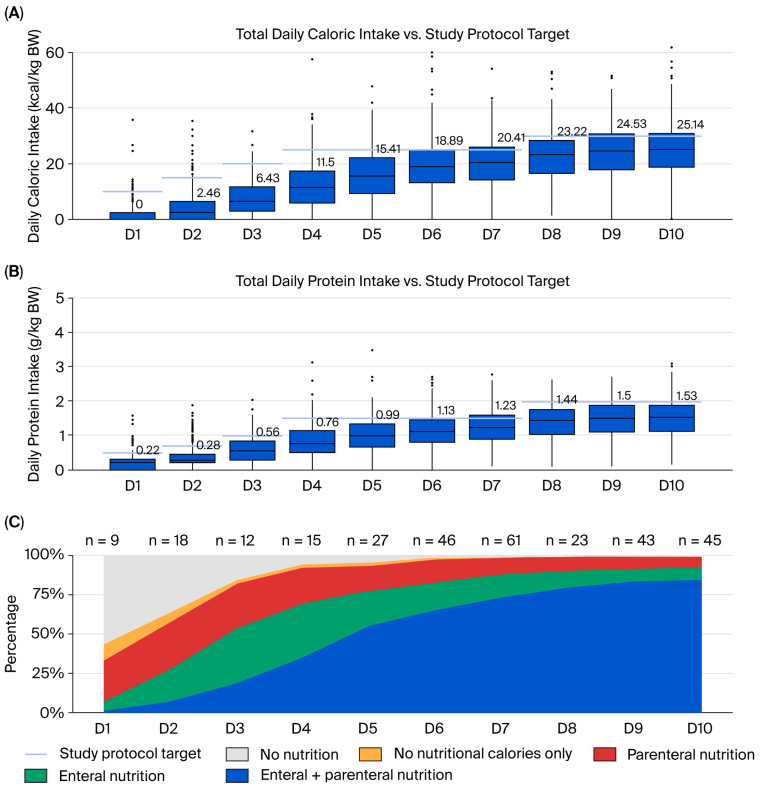
Daily nutrition delivery over the first 10 ICU days. (**A**) Boxplots of total caloric intake (kcal/kg BW/day) with medians labeled; blue horizontal segments denote the study-protocol targets of 10, 15, 20, 25, and 30 kcal/kg for days 1, 2, 3, 4–7, and 8–10, respectively. (**B**) Boxplots of protein intake (g/kg BW/day) with medians; blue segments mark targets of 0.5, 0.7, 1.0, 1.5, and 2.0 g/kg for days 1, 2, 3, 4–7, and 8–10, respectively. In both panels, medians rise over time but remain below targets, with the smallest gap by days 8–10. (**C**) Stacked area chart showing the daily distribution of nutrition routes—with per-day sample sizes (*n*) shown along the *x*-axis. Early days are dominated by no nutrition and/or PN, while EN + PN becomes predominant from about day 5 onward.

**Figure 3 nutrients-17-03180-f003:**
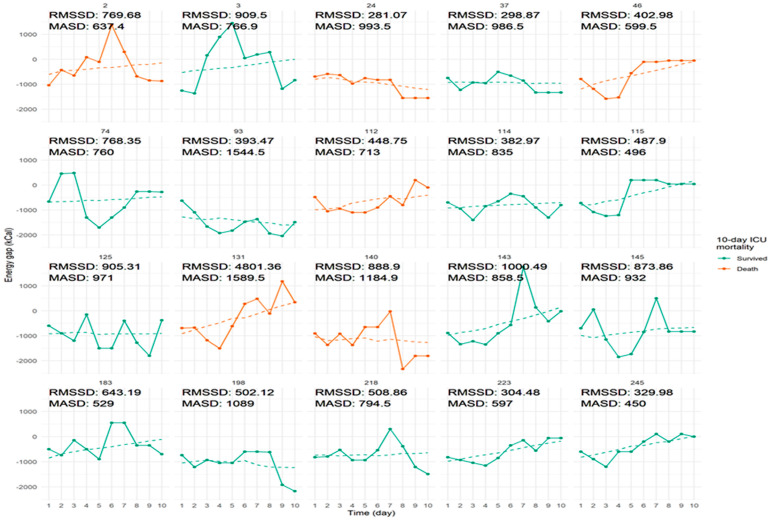
Individual energy gap trajectories stratified by 10-Day ICU mortality. Daily energy gap trajectories are shown for a sample of ICU patients over the first 10 days of admission. Each panel represents one patient, with the *x*-axis indicating ICU-day and the *y*-axis showing energy gap (kcal). Lines and points are colored by 10-day ICU outcome: green for survivors and orange for patients who died.

**Figure 4 nutrients-17-03180-f004:**
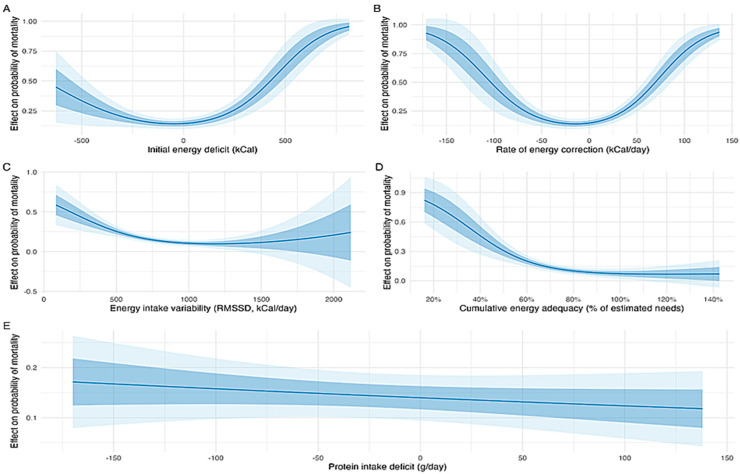
Marginal Effects of Energy and Protein Intake Dynamics on ICU Mortality. The figure showes how various energy-related factors impact mortality probability. The effects are adjusted for age, sex, APACHE II, SOFA, ISS, and CCI to control for these differences. The shaded ribbons indicate uncertainty, with wider bands representing 95% CI and narrower bands indicating ±1 standard error. These plots visualize the independent contribution of each energy-related feature on mortality probability, holding other covariates constant. (**A**) The initial energy deficit exhibits a U-shaped relationship with mortality risk; both large caloric excess and large deficits at ICU admission are associated with higher predicted mortality, with the lowest risk occurring near predicted energy targets. (**B**) The daily rate of energy correction is nonlinearly associated with outcome; moderate correction rates are linked to reduced mortality, while overly aggressive and insufficient corrections increase risk. (**C**) Very high (interruptions) or low (not adapting to patients’ needs) day-to-day variability in energy intake (RMSSD) show that both higher and lower variability correlate with greater mortality risk, though many patients had heterogeneous outcomes at extreme levels. (**D**) shows that patients receiving less than 70% of their estimated energy needs exhibit markedly increased mortality, with risk decreasing steadily as requirements approach and exceed 100%. (**E**) Deficits in protein intake are weakly associated with increased mortality probability across the observed range.

**Figure 5 nutrients-17-03180-f005:**
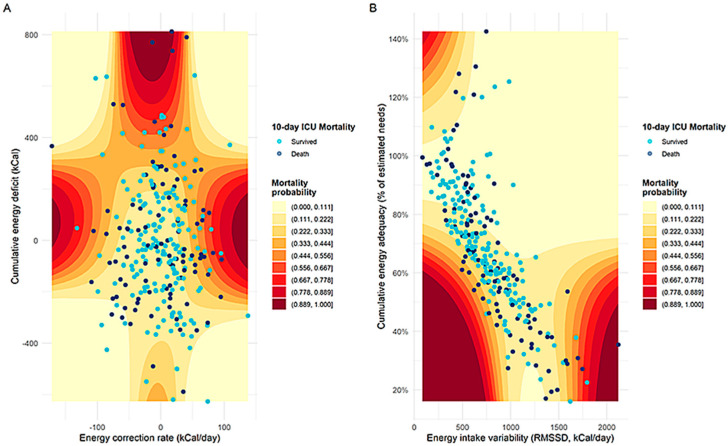
Predicted ICU Mortality Based on Energy Dynamics. The red colored zones indicate regions of high predicted mortality, with individual patient data points overlaid, reflecting 10-day ICU outcomes. Model predictions are adjusted for age, sex, APACHE II, SOFA, ISS, and CCI. (**A**): This panel displays the predicted probability of ICU mortality based on the rate of daily energy correction and baseline energy deficit, illustrating zones of high predicted mortality are those where patients had low or high demands which were not met, and those where corrections were performed too rapidly or failed to be rapid enough (i.e., interruptions). (**B**): This panel shows the predicted probability of ICU mortality based on day-to-day variability in energy intake (RMSSD) and cumulative percentage of caloric intake based on ESPEN targets, illustrating that when patients are adequately fed there is a lot more tolerability to change in nutrition before mortality risk starts to rise. Tolerability to feeding variability is shown as a wide zone of low probability of mortality (light yellow) when patients are near target and narrow zones when patients are underfed.

**Table 1 nutrients-17-03180-t001:** General and clinical characteristics of severe trauma patients (*n* = 286).

Variable	Value
Age, median (Q1, Q3)	54 (34, 71)
Male	220 (76.9)
Female	66 (23.1)
**Body Mass Index (kg/m^2^)**	
<18.5	10 (3.5)
18.5–25.5	151 (52.8)
25.5–30	78 (27.3)
>30	47 (16.4)
Charlson comorbidity index, median (Q1, Q3)	1.00 (0.00, 4.00)
**Characteristics at Admission**	
APACHE II, median (Q1, Q3)	16 (12, 20)
SAPS II, median (Q1, Q3)	45 (30, 55)
SOFA, median (Q1, Q3)	6.0 (4.0, 9.0)
GCS, median (Q1, Q3)	10.0 (7.0, 14.0)
NUTRIC, median (Q1, Q3)	3.00 (1.00, 5.00)
MV (first 48 h)	217 (75.9)
Sedation (first 48 h)	225 (78.7)
Vasopressors (first 48 h)	174 (60.8)
**Injury Characteristics**	
Blunt	241 (84.3)
Penetrating	45 (15.7)
ISS, median (Q1, Q3)	22.0 (18.0, 27.0)
**ISS Severity**	
Critical (25–75)	111 (38.8)
Severe (16–24)	175 (61.2)
Polytrauma	143 (50.0)
Damage control surgery	120 (42.0)
**Number of Surgeries**	
0	82 (28.7)
1	103 (36.0)
2	57 (19.9)
≥3	28 (9.8)
**Outcomes**	
ICU mortality	101 (35.3)
90-day mortality	101 (35.3)
ICU LOS, days, median (Q1, Q3)	15 (11, 25)
Sum days on MV, median (Q1, Q3)	10 (3, 17)
**Infections**	
Nosocomial infection	172 (60.1)
VAP	120 (41.9)
Sepsis	114 (39.8)
Septic shock	86 (30.0)
Reinfection	63 (22.0)
Primary bloodstream infection	86 (30.0)
Relapse of infection	49 (17.1)
Surgical wound infection	20 (7.0)
Intra-abdominal infection	8 (2.8)

Data are presented as *n* (%), unless otherwise stated. APACHE II, Acute Physiology and Chronic Health. Evaluation: SAPS II, Simplified Acute Physiology Score; SOFA, sepsis-related organ failure assessment; GCS, Glasgow Coma Scale; NUTRIC, nutrition risk in the critically ill; ISS, Injury Severity Score; LOS, Length of stay; MV, mechanical ventilation, VAP, Ventilator-Associated Pneumonia.

## Data Availability

The raw data supporting the conclusions of this article will be made available by the authors upon request.

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
