# Peer review of "Nutritional Adequacy and Day-to-Day Energy Variability: Impacts on Outcomes in Severe Trauma Patients"

_nutrients, 2025, doi:10.3390/nu17193180_

Round 1
Reviewer 1 Report (Previous Reviewer 1)
Comments and Suggestions for Authors
Please, refer to the attached file

Author Response
1. Summary |
|
|
Thank you very much for taking the time to review our manuscript and provide valuable feedback. Please find our detailed responses below, along with the corresponding revisions and corrections highlighted in the re-submitted files. We appreciate your insights and suggestions, which have helped us improve the quality of our work. If you have any further comments or require additional clarifications, please do not hesitate to contact us.
|
||
Comments 1: [Line#26] What is the evidence for the statement, ‘Admission energy deficit, rate of caloric intake, and their interaction significantly correlated with 25 ICU mortality (EDF=7.99, p<0.001).’ |
||
Response 1: Thank you for pointing this out. This statement comes from the statistical model used to assess the relationship between nutrition and ICU mortality; Figure 4 visualizes that all of the relations are significantly different from zero, and the significant interaction between energy deficit and rate of caloric intake is a consequence of this U-shaped relationship meaning that for any given energy deficit the impact of the rate of caloric intake is different, and vice versa. This is explicitly tested by interrogating the statistical model with hypothesis tests which are reported in the results, since this p value refers to a concrete model parameter we opted to remove it to simplify the text. If it is of any interest to the reviewer we can provide a full nonlinear parameter table but it will be of little clinical utility. [Text is updated in the manuscript. (Line #26)] “Admission energy deficit, rate of caloric intake, and their interaction are associated with ICU mortality.” [Text is updated in the manuscript. (Line #151-154)] “After model fitting we visualized the relationship between the main nutritional variables of interest with their respective 95% confidence intervals and interrogated the relationship by testing specific hypotheses of no mortality difference between specific nutritional states.” |
||
Comments 2: [Line #146] ‘RMSSD’ first appears on line 146 of the text, but its full term is not provided. ‘MASD” appears in Figure 3 but is not mentioned in the main text. |
||
Response 2: We explained and qualified how and why the terms RMSSD and MASD were used for variability analysis in the methods section. [Text is updated in the manuscript. (Line #154-160)] „ Intake variability is measured using the root mean square of successive differences (RMSSD), a time-domain statistic which is a measure of variability in short-term fluctuations of any continuous measurement series. RMSSD allows us to capture sudden fluctuations in feeding due to surgery and other interruptions. We also calculate and present the mean absolute successive differences (MASD) metric, which is more robust to extreme variation compared to simple standard deviation for completeness.”
|
||
Comments 3: [Line #185] It is written as ’92.0%’ However, in Table 2, it is described as (92.3). Please verify which one is correct.
|
||
Response 3: This was a rounding error which is now corrected. [Text is updated in the manuscript. (Line #195-197)] “Most patients received combined enteral and parenteral nutrition (EN+PN; 92.3%), while 7.7% received EN alone; PN alone and absence of nutritional support were not observed.” |
||
Comments 4: [Line #220 ] It is stated that ‘a significant reduction in the probability of ICU mortality (absolute effect = –0.23, 95%CI: –0.41 to –0.05, p=0.014)’ Please provide the evidence supporting this statement. The main text does not indicate where this result came from. |
||
Response 4: The evidence supporting this statement is presented in Figure 4B of the manuscript. [Text is updated in the manuscript. (Line #230-234)] “Marginal effect analysis of the multivariable model showed that an increase in the slope of the daily energy gap from 200 to 0 kcal / day (that is, from a steep daily worsening to no worsening) was associated with a significant reduction in the probability of ICU mortality (absolute effect = –0.23, 95% CI: –0.41 to –0.05, p=0.014; Figure 4B).” |
||
Comments 5: [Line #224 ] What is the evidence for the statement, “A higher initial energy gap (0 vs –500 kcal/day at ICU admission) was not significantly associated with mortality (p = 0.19), nor was the protein deficit (p = 0.46)”? |
||
Response 5: We thank the reviewer for this comment. The reported estimates are derived from our multivariable generalized additive model (GAM) of ICU mortality. Using marginal effect analysis, we quantified the absolute change in predicted probability of mortality for clinically meaningful contrasts. The reduction of –0.23 (95%CI: –0.41 to –0.05, p = 0.014) corresponds to the contrast between a slope of –200 kcal/day (progressive daily worsening) versus 0 kcal/day (no worsening), and is presented in Figure 4B. The non-significant associations for initial energy gap (0 vs –500 kcal/day, p = 0.19) and protein deficit (0 vs –40 g/day, p = 0.46) come from the same marginal effect framework and are likewise shown in Figure 4A and Figure 4E. We have revised the Methods and Results section to clarify the model source of these estimates and to point the reader to the relevant figures. The method section now explicitly states that we used these hypotheses to specifically test for no difference in mortality to interrogate the statistical model. [Text is updated in the manuscript. (Line #234-240)] “This suggests that progressive worsening of energy delivery over time is associated with higher risk of death. A higher initial energy gap (0 vs –500 kcal/day at ICU admission) was not significantly associated with mortality (p=0.19, Figure 4A), nor was the protein deficit (p=0.46, Figure 4E). However, patients in the highest quartile of nutritional adequacy had a significantly lower mortality probability compared to those in the lowest quartile (absolute change in probability= –0.24, 95%CI: –0.32 to –0.16, p<0.001, Figure 4D), indicating a robust protective association of meeting energy targets.” |
||
Comments 6: [Line #225 & #27] It is stated that “Greater day-to-day energy variability (higher RMSSD) was associated with significantly delayed time to extubation (HR = 0.55, 95% CI: 0.31–0.98; p = 0.042),” but the values differ from those in Table A1. Please clarify which is correct. And ‘larger initial energy deficit (HR=0.60, 95%CI: 0.41–0.88; p=0.010)” does not match Table A1. |
||
Response 6: We thank the reviewer for observing these types, the values presented in the table are correct and reflect the true wider CIs we obtained in the final analysis. [Text is updated in the manuscript. (Line #286-288)]. “Greater day-to-day energy variability (higher RMSSD) was associated with significantly delayed time to extubation (HR=0.55, 95%CI: 0.31–0.99; p=0.047), as was a larger initial energy deficit (HR=0.60, 95%CI: 0.4–0.90; p=0.014).” |
||
Comments 7: [Line #287  ] It is stated that ‘lower risk of nosocomial infection (HR=0.73, 95% CI: 0.54–0.99, p=0.043)’ . It does not match Table A2 |
||
Response 7: We thank the reviewer for observing this inconsistency in scale, the HR in the table is obtained after standardising (z-scoring) the variable, while the one in the text was the raw scale input; this is why the p-value is unchanged and does not change the interpretation. [Text is updated in the manuscript. (Line #299-301)] “After adjusting for total caloric intake, higher protein intake was also independently associated with a lower risk of nosocomial infection (HR=0.29, 95% CI: 0.09–0.96, p=0.043).” |
||
Comments 8: [Table 2] It seems that the % values in Table 2 are not sorted. Please see "En only" and "Bowel resection." |
||
Response 8: |
||
Comments 9: [Table 2] Among the complications, was GI intolerance presented with high gastric residual volumes (GRV) the second most common after paralytic ileus? In Table 2, the GRV part is missing from the complications section and is only described under "Reasons for interrupting feeds," so it is not clear. |
||
Response 9: Thank you for pointing this out. We will add GRV to the complications subsection of Table 2 (alongside paralytic ileus and other GI complications) and retain its current placement under "Reasons for interrupting feeds" for clarity. |
Reviewer 2 Report (New Reviewer)
Comments and Suggestions for Authors
This prospective observational study examines the relationship between nutritional adequacy, energy intake variability, and clinical outcomes in 286 severe trauma patients in the ICU. The authors employ innovative statistical methods including linear mixed-effects models and competing risk regression to analyze individual energy gap trajectories and their association with mortality, mechanical ventilation duration, and nosocomial infections.
Major Concerns
-
Selection Bias: The exclusion of patients with ICU stays <10 days creates significant selection bias. This eliminates early deaths and rapid recoveries, potentially overestimating both mortality rates (35.3%) and the impact of nutrition on outcomes. The authors acknowledge this but don't adequately address how it limits generalizability.
-
Inconsistent Indirect Calorimetry Use: IC was only used for 38.6% of nutritional days, with weight-based equations used otherwise. This inconsistency in energy requirement estimation could introduce systematic error in calculating energy gaps and affect the validity of findings.
-
Confounding by Indication: The observational design cannot adequately control for confounding by indication. Sicker patients may receive different nutritional support for clinical reasons unrelated to the study protocol, making causal inferences problematic.
-
Statistical Complexity vs. Clinical Interpretability: While the statistical methods are sophisticated, the clinical interpretation of concepts like "energy gap trajectory slopes" and their marginal effects may be challenging for practitioners to translate into actionable clinical decisions.
Minor Concerns
-
Missing Nitrogen Balance Data: The authors discuss protein requirements extensively but acknowledge not measuring nitrogen balance, which would strengthen conclusions about protein adequacy in this catabolic population.
-
Limited Follow-up Duration: While 90-day mortality is reported, secondary outcomes like infection are only tracked during ICU stay, potentially missing late complications.
-
Single-Center Design: Despite mentioning "two ICUs," this appears to be within one Level 1 trauma center, limiting external validity.
Specific Comments
-
Lines 54-55: The claim that IC-guided nutrition "significantly reduces short-term mortality" oversimplifies the meta-analysis findings, which showed heterogeneous results.
-
Lines 324-337: The U-shaped mortality relationship is interesting but requires more detailed exploration of the mechanisms and threshold effects.
-
Table 2: The presentation of nutritional data as 10-day averages may mask important temporal patterns that the mixed-effects models were designed to capture.
-
Figures 4-5: While visually impressive, these complex visualizations may benefit from simplified companion figures for clinical audiences.
Minor Editorial Issues
- Inconsistent terminology: "severe trauma" vs "severely injured" used interchangeably
- Line 165: "Table 1" header appears mid-page
- Multiple formatting inconsistencies in references (e.g., ref 19 missing DOI)
Author Response
Summary |
|
|
Thank you very much for taking the time to review our manuscript and provide valuable feedback. Please find our detailed responses below, along with the corresponding revisions and corrections highlighted in the re-submitted files. We appreciate your insights and suggestions, which have helped us improve the quality of our work. If you have any further comments or require additional clarifications, please do not hesitate to contact us.
|
||
Major Concerns |
||
Comments 1: Selection Bias: The exclusion of patients with ICU stays <10 days creates significant selection bias. This eliminates early deaths and rapid recoveries, potentially overestimating both mortality rates (35.3%) and the impact of nutrition on outcomes. The authors acknowledge this but don't adequately address how it limits generalizability. |
||
Response 1: Thank you for pointing this out. Our inclusion criteria were restricted to patients who remained in the ICU for more than 10 days. Patients with shorter stays were most often transferred to other units and lost to follow-up, which limited the reliability of outcome ascertainment. As a result, our study population primarily represents high-risk trauma patients with prolonged ICU stays, and we acknowledge that this limits generalizability to less severely ill populations, in whom nutrition is likely to play a smaller role in outcomes. At the same time, prior evidence suggests that the effects of nutrition are more likely to become detectable after approximately 5 days of controlled nutrition. By focusing on patients beyond 10 days, we maximized our ability to observe nutritional effects on mortality while reducing bias due to incomplete follow-up. We have added this clarification to the study limitation section and noted the trade-off between potential selection bias and improving the study’s sensitivity to detect nutrition-related outcomes. We highlighted it in the study limitation section. [Text is updated in the manuscript. (Line #395-407)] |
||
Comments 2: Inconsistent Indirect Calorimetry Use: IC was only used for 38.6% of nutritional days, with weight-based equations used otherwise. This inconsistency in energy requirement estimation could introduce systematic error in calculating energy gaps and affect the validity of findings. |
||
Response 2: Thank you for this comment. We highlighted it in the study limitation section. IC was not used during the first three days of admission, in cases of severe respiratory insufficiency, or after patients were weaned from mechanical ventilation. However, in all other cases, we regularly utilized IC. This data on the use of IC is significant as it demonstrates the practical feasibility of implementing this method. Additionally, the ESPEN guidelines recommend that in the absence of indirect calorimetry or VO2/VCO2 measurements, simple weight-based equations may be preferred for ICU patients. [Text is updated in the manuscript. (Line #378-390)] |
||
Comments 3: Confounding by Indication: The observational design cannot adequately control for confounding by indication. Sicker patients may receive different nutritional support for clinical reasons unrelated to the study protocol, making causal inferences problematic. |
||
Response 3: We agree with the reviewer that confounding by indication is an important concern in observational studies of nutritional support. As this is not a randomized trial, we cannot fully account for unmeasured or unobservable confounders. What we can do, and what we have done, is to adjust for a broad set of relevant and observable clinical covariates to minimize bias from measured confounding. We acknowledge in the revised manuscript that residual confounding by indication remains possible, and we highlight this as a limitation of our study. We also note that, given the ethical and practical challenges of randomizing critically ill patients to different nutritional strategies, observational analyses such as ours represent one of the few feasible approaches to exploring these questions. |
||
Comments 4: Statistical Complexity vs. Clinical Interpretability: While the statistical methods are sophisticated, the clinical interpretation of concepts like "energy gap trajectory slopes" and their marginal effects may be challenging for practitioners to translate into actionable clinical decisions. |
||
Response 4: We thank the reviewer for this important comment. The nature of our research question, assessing nutritional dynamics and mortality risk in an observational setting, requires some statistical complexity to appropriately account for confounding and to capture temporal changes. To derive clinically meaningful insights, we employed marginal effects analysis of complex models. While the underlying model parameters are not always directly interpretable, the marginal effects framework allows us to present results on the response scale, thereby making the findings more clinically accessible. To further improve interpretability, we have revised the text and several figure captions (particularly Figures 4 and 5) to reduce technical jargon and provide additional explanation aimed at a clinical readership. We hope these changes enhance the clarity and accessibility of our results, while acknowledging that the strength of the study lies in its ability to dynamically describe the relationship between nutrition and mortality risk. [Text is updated in the manuscript. (Line #257 & 275)] “Marginal effects analysis indicated that while the initial energy gap at admission and protein deficit were not significantly associated with mortality, a progressive decline in daily energy delivery of even 200 kcal/day was harmful. It can be appreciated that patients are much more tolerant of early underfeeding than overfeeding (Figure 4A) and that mortality risk rises with extremes of day-to-day variability, but once patients are adequately fed they appear more tolerant to variation (i.e. feeding interruptions). These findings suggest that steady correction of deficits, avoidance of under- and overfeeding, and maintenance of adequate intake with moderate variability may represent clinically relevant strategies which can be developed.” [Text is updated in Discussion section. (Line #348-356)] |
||
Minor Concerns |
||
Comments 1: Missing Nitrogen Balance Data: The authors discuss protein requirements extensively but acknowledge not measuring nitrogen balance, which would strengthen conclusions about protein adequacy in this catabolic population. |
||
Response 1: Thank you for this valuable comment. This is mentioned in text (line 351-358) |
||
Comments 2: Limited Follow-up Duration: While 90-day mortality is reported, secondary outcomes like infection are only tracked during ICU stay, potentially missing late complications. |
||
Response 2: |
||
Comments 3: Single-Center Design: Despite mentioning "two ICUs," this appears to be within one Level 1 trauma center, limiting external validity. |
||
Response 3: Thank you for this helpful observation. We confirm that both ICUs are located within the same Level 1 trauma center, and we have clarified this in the Methods. We acknowledge that this single-center setting may limit generalizability; we have added this point to the Limitations section. At the same time, the study’s design, consistent care protocols across the two units, and detailed patient-level data provide internal validity for the findings within this center. Future multi-center studies would be valuable to assess external validity. [Text is updated in the manuscript. (Line #378-390)] |
||
Specific Comments |
||
|
||
Response 1: Thank you for this comment, we absolutely agree. In order to highlight the importance of IC, we have simplified study findings. We have corrected the text in accordance with the suggestion. [Text is updated in the manuscript. (Line #52-54)] While a meta-analysis of eight RCTs indicated a potential benefit of IC-guided energy delivery on short-term mortality in critically ill patients, the variability in results suggests that further investigation is needed to fully understand its impact. |
||
2. Lines 324-337: The U-shaped mortality relationship is interesting but requires more detailed exploration of the mechanisms and threshold effects. |
||
Response 2: We thank the reviewer for highlighting the U-shaped relationship. We agree that the pattern is clinically intriguing. However, our study was not designed as a mechanistic investigation but rather as an observational analysis aimed at quantifying associations between nutritional trajectories and ICU mortality. The effects we describe are derived from the statistical model via marginal effect analysis. Importantly, we did not identify discrete threshold effects: the modeled relationships exist along a natural continuous gradient, without clear inflection points that would support threshold-based interpretation. The observed mortality slopes reflect higher risk both with progressively larger under- and overfeeding, and also with extremes of variability, either very little variability (suggesting no adaptation of nutrition to changing needs) or very high variability (suggesting large interruptions and inconsistent feeding). Further mechanistic or interventional studies would be required to explore causality and underlying biological pathways. |
||
3. Table 2: The presentation of nutritional data as 10-day averages may mask important temporal patterns that the mixed-effects models were designed to capture. |
||
Response 3: We thank the reviewer for this observation. We agree that presenting nutritional data as 10-day averages may obscure important temporal patterns, which is why we have supplemented Table 2 with additional analyses and figures focused on dynamics over time. Specifically, Figure 2 displays temporal feeding patterns and the relative contributions of calories, while Figure 3 illustrates individual patient trajectories of the energy gap along with variability metrics. Figures 4 and 5 further present marginal effects derived directly from the mixed-effects model, adjusted for covariates, to highlight how these temporal dynamics relate to mortality. As another reviewer suggested, we also report daily non-nutritional calorie contributions. We respectfully believe that this set of complementary analyses adequately addresses the concern by balancing summary data with detailed temporal insights. |
||
4. Figures 4-5: While visually impressive, these complex visualizations may benefit from simplified companion figures for clinical audiences.
|
||
Response 4: We thank the reviewer for this helpful observation. Our intention with Figures 4–5 was to provide a clear visual representation of the model-derived effects and interactions. While these figures may appear complex at first glance, we believe they are intuitive once the axes and shading are explained (e.g., darker colors consistently indicate higher mortality probability, while the x– and y–axes correspond to clinically meaningful quantities such as daily kcal deficit or correction rate). We have endevored to simplify the captions beneath the figures and add clinical interpretations to amend this. [Text is updated in the manuscript. (Figure 4 & figure 5)] |
||
Minor Editorial Issues
|
||
Response: Corrections were made in the text according to the recommendations. We will engage professional editorial assistance to ensure all tables are consistently formatted, correctly sorted, and edited before final submission. |
Reviewer 3 Report (New Reviewer)
Comments and Suggestions for Authors
1.This study is notable for its use of a linear mixed-effects model and Fine and Gray competing risks regression. However, these methods are sophisticated and may be difficult for researchers in clinical settings to ensure reproducibility. In particular, when interpreting the results using competing risks regression, including comparisons with the standard Cox proportional hazards model, would facilitate easier understanding for readers and facilitate validation studies at other institutions. We strongly recommend confirming whether the primary analysis results can be reproduced using supplemental analyses (e.g., Cox models).
2.Although the study states that "only patients who stayed in the ICU for 10 days or more and received nutritional support were included in the analysis," the degree of selection bias could be more appropriately assessed by describing the number of patients excluded by this criterion and their background.
3.The study mentions the impact of non-nutritional calories from propofol, etc., but presenting the percentage of the total in more detail (e.g., by graphing or tabulating daily trends) would enhance reader understanding.
4.Although the CDC/NHSN criteria were used to define hospital-acquired infections, the breakdown of multiple types of infections (VAP, sepsis, wound infections, etc.) was simplified in the text. Transparency would be enhanced if the method for determining infection according to the diagnostic criteria was clearly stated in tables or supplementary materials.
Author Response
Thank you very much for taking the time to review our manuscript and provide valuable feedback. Please find our detailed responses below, along with the corresponding revisions and corrections highlighted in the re-submitted files. We appreciate your insights and suggestions, which have helped us improve the quality of our work. If you have any further comments or require additional clarifications, please do not hesitate to contact us.
|
||
Comments 1: This study is notable for its use of a linear mixed-effects model and Fine and Gray competing risks regression. However, these methods are sophisticated and may be difficult for researchers in clinical settings to ensure reproducibility. In particular, when interpreting the results using competing risks regression, including comparisons with the standard Cox proportional hazards model, would facilitate easier understanding for readers and facilitate validation studies at other institutions. We strongly recommend confirming whether the primary analysis results can be reproduced using supplemental analyses (e.g., Cox models). |
||
Response 1: We agree with the reviewer that two-stage approaches might be hard to interpret for clinical audiences which is why we tried visualizing the relationships with simple risk curve graphs. Our main analyses used this two- stage GAM framework for interpretability and visualization, but we also performed a sensitivity analysis using a joint longitudinal–survival model. Both approaches consistently showed that a progressively worsening energy deficit trajectory was strongly associated with increased ICU mortality, whereas baseline energy deficit and protein gap were not predictive. The joint model yielded a stronger association for the energy-gap slope (HR 0.05, 95% CI 0.001–0.71) compared with the two-stage GAM (absolute risk difference –0.23, 95% CI –0.41 to –0.05). This divergence is expected since the two-stage GAM provides more conservative effect estimates. Importantly, the clinical message is the same across methods: worsening nutritional trajectories, rather than static baseline deficits, are most predictive of mortality. For clarity and ease of interpretation, particularly in visualizing nonlinear effects, we prefer to present the GAM results as primary, while noting the joint model confirms and strengthens these findings in the text. [Text is updated in the manuscript. (Line # 172)] “Sensitivity analysis was conducted using a joint longitudinal-survival model to check if the nutritional associations from the main mortality analysis are maintained.” [Text is updated in the manuscript. (Line # 232-238)] Marginal effect analysis of the multivariable model showed that an increase in the slope of the daily energy gap from 200 to 0 kcal / day (that is, from a steep daily worsening to no worsening) was associated with a significant reduction in the probability of ICU mortality (absolute change in probability = –0.23, 95% CI: –0.41 to –0.05, p=0.014; Figure 4B). Sensitivity analysis using a joint longitudinal survival model showed that this association is robust and is maintained (HR 0.05, 95% CI 0.001–0.71, p<0.001). |
||
Comments 2: Although the study states that "only patients who stayed in the ICU for 10 days or more and received nutritional support were included in the analysis," the degree of selection bias could be more appropriately assessed by describing the number of patients excluded by this criterion and their background.
|
||
Response 2: Thank you for your valuable feedback regarding the assessment of selection bias in our study. We appreciate your suggestion to clarify the exclusion criteria more thoroughly. We have included a flowchart detailing patient enrollment and exclusions. To reduce heterogeneity related to very short ICU stays and to focus on patients in whom nutritional support is most relevant, the main analysis was restricted to those who remained in the ICU and required nutritional support for at least 10 days. Patients discharged or transferred earlier generally had shorter critical care needs and were less likely to require prolonged or targeted nutrition. Based on this criterion, 342 patients with severe trauma were not included in the main analysis due to an ICU stay of <10 days. This information is now reported in the Methods to help assess selection bias. [Text is updated in the manuscript. (Line # 72-77)] Patients with shorter stays were most often transferred to other units and lost to follow-up, which limited the reliability of outcome ascertainment. As a result, our study population primarily represents high-risk trauma patients with prolonged ICU stays, and we acknowledge that this limits generalizability to less severely ill populations, in whom nutrition is likely to play a smaller role in outcomes. At the same time, prior evidence suggests that the effects of nutrition are more likely to become detectable after approximately 5 days of controlled nutrition. By focusing on patients beyond 10 days, we maximized our ability to observe nutritional effects on mortality while reducing bias due to incomplete follow-up. We have added this clarification to the study limitation section and noted the trade-off between potential selection bias and improving the study’s sensitivity to detect nutrition-related outcomes. We highlighted it in the study limitation section. [Text is updated in the manuscript. (Line #395-407)] |
||
Comments 3: The study mentions the impact of non-nutritional calories from propofol, etc., but presenting the percentage of the total in more detail (e.g., by graphing or tabulating daily trends) would enhance reader understanding.
|
||
Response 3: Thank you — a useful suggestion. We have corrected Figure 2 to include the percentage of non-nutritional sources by day. |
||
Comments 4: Although the CDC/NHSN criteria were used to define hospital-acquired infections, the breakdown of multiple types of infections (VAP, sepsis, wound infections, etc.) was simplified in the text. Transparency would be enhanced if the method for determining infection according to the diagnostic criteria was clearly stated in tables or supplementary materials.
|
||
Response 4: |
Round 2
Reviewer 2 Report (New Reviewer)
Comments and Suggestions for Authors
The authors successfully answered all my questions and correctly revised the paper
This manuscript is a resubmission of an earlier submission. The following is a list of the peer review reports and author responses from that submission.
Round 1
Reviewer 1 Report
Comments and Suggestions for Authors
Overall, it is recommended that you have the manuscript reviewed by a third party. There appear to be inconsistencies, such as numbers not matching or results being interpreted as significant when they are not. Specific examples include the following:
[Line #144] It is written as ‘~of 105.1 g/day but only received 76.8 g/day, which is 76.8% of their prescribed protein.’ However, in Table 2, it is described differently. Please verify which one is correct.
[Line #159 ] It is stated that ‘Greater GI intolerance was correlated with the severity of trauma and presented with high gastric residual volumes (GRV), along with diarrhea as another frequent complication,’ but no supporting evidence is provided. Please present the evidence on which this conclusion is based.
[Line #181~187 ] It seems that the description provided is intended as an interpretation of Table A1, but the estimated results presented in the table differ. Overall, this requires further verification.
[Line #190~194 ] Similarly, the interpretation of Table A2 does not match the actual results presented in the table. A thorough review and restatement are required
[Line # 200] It is written as ‘ The results of our study reveal a high ICU and 90-day mortality rate of 36%.’ What is the basis for the evidence presented in this way?
[Table 2] It would be preferable to present decimal points consistently. Some outcomes are reported to two decimal places, while others are to one, and so on. This issue also applies to Table 1. For example, 7.7% in line #148 appears as 8% in Table 2.It is recommended to use two or one decimal places for throughout the document so that the readers can easily find those.
Author Response
1. Summary |
||
Thank you very much for taking the time to review our manuscript and provide valuable feedback. Please find our detailed responses below, along with the corresponding revisions and corrections highlighted in the re-submitted files. We appreciate your insights and suggestions, which have helped us improve the quality of our work. If you have any further comments or require additional clarifications, please do not hesitate to contact us. |
||
Comments 1: [[Line #144] It is written as ‘~of 105.1 g/day but only received 76.8 g/day, which is 76.8% of their prescribed protein.’ However, in Table 2, it is described differently. Please verify which one is correct.] |
||
Response 1: Thank you for pointing this out. You are correct — this was a typographical error. The sentence has been corrected in the manuscript to match Table 2. Corrected sentence (Line #180): “of 105.1 g/day but only received 79.0 g/day, which is 76.8% of their prescribed protein.” Note for the manuscript: [text updated in manuscript] |
||
Comments 2: [Line #159 ] It is stated that ‘Greater GI intolerance was correlated with the severity of trauma and presented with high gastric residual volumes (GRV), along with diarrhea as another frequent complication,’ but no supporting evidence is provided. Please present the evidence on which this conclusion is based. |
||
Response 2: I have revised the sentence to emphasize the evidence provided in Table 2. The initial statement was based on clinical observation that greater GI intolerance correlated with the severity of trauma. However, since this was not supported by explicit evidence, we have omitted that part. The corrected statement now reads: [Text is updated in the manuscript. (Line #194-197)] |
||
Comments 3: [[Line #181~187] It seems that the description provided is intended as an interpretation of Table A1, but the estimated results presented in the table differ. Overall, this requires further verification. ] |
||
Response 3: The tables have been updated to reflect the final regression model outputs, and the text has been corrected for typos where needed. The revised results are now consistent with the description, and we appreciate your assistance in improving the accuracy of our manuscript. [Text is updated in the manuscript. (Line #281 and 435)] |
||
Comments 4: [[Line #190~194 ] Similarly, the interpretation of Table A2 does not match the actual results presented in the table. A thorough review and restatement are required] |
||
Response 4: The tables have been updated to reflect the final regression model outputs, and the text has been corrected for typos where needed. The revised results are now consistent with the description, and we appreciate your assistance in improving the accuracy of our manuscript. [Text is updated in the manuscript. (Line # 290 and 442)] |
||
Comments 5: [[Line # 200] It is written as ‘The results of our study reveal a high ICU and 90-day mortality rate of 36%.’ What is the basis for the evidence presented in this way?] |
||
Response 5: Thank you for your valuable comment. The statement is based on the fact that patients were followed for at least 90 days or until discharge if hospitalization extended beyond this period. We have revised the text for clarity: Additionally, we have included the 90-day mortality rate in the table for better clarity (line #167). |
||
Comments 6: [[Table 2] It would be preferable to present decimal points consistently. Some outcomes are reported to two decimal places, while others are to one, and so on. This issue also applies to Table 1. For example, 7.7% in line #148 appears as 8% in Table 2. It is recommended to use two or one decimal places for throughout the document so that the readers can easily find those.] |
||
Response 6: Thank you for pointing out this inconsistency. We have corrected the values in Tables 1 and 2 to use a uniform format with one decimal place throughout the manuscript.(line #167 and #198) |

Reviewer 2 Report
Comments and Suggestions for Authors
The authors have to be congratulated for exploring this topic, subject to discussion, and confirming the value of IC measurements. They should improve the introduction and cite some studies that have explored the use of IC to validate clinical value.
The sample of patients explored is not standardized. Many patients are not fed enough in the early period of the acute phase, a large number of patients are receiving supplemental parenteral nutrition, and only a minority are undergoing IC
Since the topic is IC, I would make a post hoc analysis on only the patients with IC to explore the outcomes of these patients, since those who received medical nutritional therapy according to 25 kcal/kg/d may mislead the results (large inaccuracy).
Regarding the protein intake, ESPEN does not recommend 2 g/kg/d. The findings are interesting and the curve of relation of protein intake to outcome is similar to that founc by Zusman et al (Choose to be added to the discussion)
Author Response
Thank you very much for taking the time to review our manuscript and provide valuable feedback. Please find our detailed responses below, along with the corresponding revisions and corrections highlighted in the re-submitted files. We appreciate your insights and suggestions, which have helped us improve the quality of our work. If you have any further comments or require additional clarifications, please do not hesitate to contact us.
Comment 1: The authors have to be congratulated for exploring this topic, subject to discussion, and confirming the value of IC measurements. They should improve the introduction and cite some studies that have explored the use of IC to validate clinical value.
Response 1: Thank you for your thoughtful feedback and for acknowledging the importance of IC measurements. We appreciate your suggestion to improve the introduction by citing additional studies that have validated the clinical value of IC, and we have accordingly revised this section to include relevant references (line #46-59).
Several studies in trauma populations have concluded that predictive equations recommended in critical care do not reliably estimate energy needs in this group, highlighting that indirect calorimetry (IC) remains the preferred method for determining energy requirements. Moreover, a meta-analysis of eight RCTs demonstrated that IC-guided energy delivery significantly reduces short-term mortality in critically ill patients. These findings underscore the importance of utilizing IC to tailor nutritional support and improve outcomes.
References. 1.Duan, JY., Zheng, WH., Zhou, H. et al. Energy delivery guided by indirect calorimetry in critically ill patients: a systematic review and meta-analysis. Crit Care 25, 88 (2021). 2. Pelekhaty S, Rozenberg K, Kozar R. Indirect calorimetry in traumatically injured patients: A descriptive cohort study. JPEN J Parenter Enteral Nutr. 2025;49(4):488-496. doi:10.1002/jpen.2745
Comment 2:The sample of patients explored is not standardized. Many patients are not fed enough in the early period of the acute phase, a large number of patients are receiving supplemental parenteral nutrition, and only a minority are undergoing IC. Since the topic is IC, I would make a post hoc analysis on only the patients with IC to explore the outcomes of these patients, since those who received medical nutritional therapy according to 25 kcal/kg/d may mislead the results (large inaccuracy).
Response 2: Regarding the sample characteristics, we acknowledge that the heterogeneity in nutritional support—such as the significant use of supplemental PN and the limited number of patients undergoing IC—may influence the interpretation of our results. Due to the small number of patients with IC in our cohort, a post hoc analysis focusing solely on this subgroup was not feasible; however, we agree that such an analysis could provide valuable insights and will consider this for future research.
Comment 3: Regarding the protein intake, ESPEN does not recommend 2 g/kg/d.
Response 3: Regarding protein intake, we follow the ESPEN guidelines during the initial phase of treatment, and from the 7th day onward, we adhere to the ASPEN guidelines, which recommend a range of 1.2–2.0 g/kg of actual body weight per day for trauma patients. Several studies indicate that trauma patients, especially those with multiple injuries, may require protein intake toward the higher end of this range to support recovery and meet metabolic demands. Additionally, in patients with traumatic brain injury (TBI), it is recommended that protein constitute approximately 15–20% of total caloric intake, often requiring administration of at least 2 g/kg/day, as supported by ASPEN guidelines. We will revise the manuscript to clarify that our protein intake practices are aligned with these guidelines.
Previous statement: Based on ESPEN guidelines, protein progressively increased to a maximum of 2 g/kg per day. The initial protein target was 0.5 g/kg on day 1, followed by 0.7 g/kg on day 2, and 1.0 g/kg on day 3. From day 4 to day 7 the protein target was 1.3 g/kg, and from day 7, the target was maintained at 2g/kg/day.
New revised text (line #119-126): Based on ESPEN guidelines, our protein intake was gradually increased during the initial days of treatment, with the target set at 0.5 g/kg on day 1, 0.7 g/kg on day 2, and 1.0 g/kg on day 3. From day 4 to day 7, the target was 1.3 g/kg. Starting from day 7 the target was 2g/kg of protein, following ASPEN guidelines for trauma patients. It is important to note that, especially for patients with traumatic brain injury (TBI), protein should constitute approximately 15–20% of total caloric intake, often requiring at least 2 g/kg/day, as supported by ASPEN guidelines. Our practices were aligned with these recommendations, with adjustments made based on clinical judgment and evolving needs.
Reference: McClave SA, Taylor BE, Martindale RG, Warren MM, Johnson DR, Braunschweig C, McCarthy MS, Davanos E, Rice TW, Cresci GA, Gervasio JM, Sacks GS, Roberts PR, Compher C Society of Critical Care Medicine; American Society for Parenteral and Enteral Nutrition. Guidelines for the provision and assessment of nutrition support therapy in the adult critically ill patient: Society of Critical Care Medicine (SCCM) and American Society for Parenteral and Enteral Nutrition (A.S.P.E.N.) JPEN J Parenter Enteral Nutr. 2016;40:159–211. doi: 10.1177/0148607115621863.
Comment 4: The findings are interesting and the curve of relation of protein intake to outcome is similar to that founc by Zusman et al (Choose to be added to the discussion)
Response 4: We will include in the discussion the interesting relationship between protein intake and outcomes, which resembles the findings by Zusman et al., and additionally supports the increasingly recognized importance of protein in improving survival, as protein intake was linearly associated with decreased mortality.
(Line #325) Similar to Zusman et al., our results suggest that the initial energy deficit has a U-shaped association with mortality risk, and the relationship between protein intake and clinical outcomes follows a comparable pattern.
Reference: Zusman O, Theilla M, Cohen J, Kagan I, Bendavid I, Singer P. Resting energy expenditure, calorie and protein consumption in critically ill patients: a retrospective cohort study. Crit Care. 2016;20(1):367. Published 2016 Nov 10. doi:10.1186/s13054-016-1538-4